# Implications for Precision Accelerated Clinically Embedded Research (PACER): A novel technology-enabled approach to conducting minimal-risk research in outpatient community healthcare settings

Emma Friedman[1,2☯], Kelly Nicole Michelson[1,2☯*], Shruti Sehgal[1], Russell Steans[3], Mohammad Hosseini[1], Matthew J. Baumann[1], Amanda K. Venables[1], Theresa L. Walunas[1], Justin Starren[1,4]

1 Northwestern University Feinberg School of Medicine, Chicago, Illinois, United States of America, 2 Ann & Robert H Lurie Children's Hospital of Chicago, Chicago, Illinois, United States of America, 3 National Institutes of Health, Bethesda, Maryland, United States of America, 4 Department of Medical Imagining College of Medicine, University of Arizona, Tucson, Arizona, United States of America

☯ These authors contributed equally to this work.

* kmichelson@luriechildrens.org

## Abstract

Current challenges in the clinical research landscape include insufficient enrollment of study participants, lack of study participant diversity, protracted study progression, and the siloing of research within academic medical centers. Recent advances in technology could minimize barriers to producing effective, timely, and comprehensive clinical research by addressing issues from study design to dissemination of results. Particularly, the Fast Health Interoperability Resources standards and Clinical Decision Support Hooks could support data acquisition, sharing, and expansion of research across organizations and disparate electronic health records. We developed a novel approach, Precision Accelerated Clinically Embedded Research (PACER), that leverages these advances in healthcare technology to integrate very short, minimal-risk research activities into clinical encounters. PACER could enable scalable, efficient, and cost-effective clinical research and has enormous potential. However, PACER also presents potential ethical, sociotechnical, and implementation quandaries. The current study aimed to obtain insights on these matters from relevant individuals. We conducted 47 qualitative semi-structured interviews with patients, clinicians, research experts (individuals involved in developing and conducting research), and bioethicists. We sought participants' perspectives on the potential ethical, sociotechnical, and implementation issues raised by PACER. We identified five key domains: impacts on clinical research, consent, compensation, impacts on people and organizations, and implementation. We examined interview participants' views using bioethical principles of autonomy, justice/fairness, beneficence, and nonmaleficence. While participants had diverse views, these insights highlight important considerations for PACER implementation and suggest areas for future empirical work.

**Data availability statement:** Data are available in the ICPSR submission number: ICPSR-21612, DOI: https://doi.org/10.3886/E219461V.

**Funding:** This work was funded by The Greenwall Foundation URL: https://greenwall.org/. and also by the National Institutes of Health's National Center for Advancing Translational Sciences, Grant Number: UL1TR001422. The study funders did not play any role in the study design, data collection and analysis, decision to publish, or preparation of the manuscript.

**Competing interests:** Aside from those listed below, the authors attest that that no other grants, funding sources, or affiliations are related to, interfere with, or could reasonably be perceived as interfering with the full and objective presentation, peer review, editorial decision-making, or publication of this particular research study. 1. M.B. received salary support funding from National Evaluation Center (NEC) for the Network of the National Library of Medicine (NNLM) Grant number: 5U24LM013751-04 URL: https://urldefense.com/v3/__https://www.nnlm.gov/funding__;!!M0gZt9BicGig!iccQX-3d6AyaiU94J9kCEmt3ysPlVA0viXfZOYZ_aIb7aiJfFfp08B-Zr1CvvleU7MkbFq7nBtbs6ycT97dsYTjs$ 2. M.H. was supported by a grant from National Institutes of Health's National Center for Advancing Translational Sciences Grant number: UM1TR005121 URL: https://urldefense.com/v3/__https://ncats.nih.gov/funding/overview__;!!M0gZt9BicGig!iccQX3d6AyaiU94J9kCEmt3ysPlVA0viXfZOYZ_aIb7aiJfFfp08B-Zr1CvvleU7MkbFq7nBtbs6ycT9s4vnqH8$ The work of E.F., K.M. and J.S. was supported by funding from The Greenwall Foundation URL: https://greenwall.org/. J.S. was supported by the National Institutes of Health's National Center for Advancing Translational Sciences, Grant Number: UL1TR001422, None of the funders played any role in the study design, data collection and analysis, decision to publish, or preparation of the manuscript for any of the authors. This work does not represent the views of the NIH, NCATS or US government.

## Introduction

The mission of Precision Accelerated Clinically Embedded Research (PACER) is to maximize opportunities for broad and diverse participation in clinical research by expanding access to research studies and thereby improving the generalizability of results. This goal is shared by the National Institutes of Health, among other research organizations. The NIH's newly appointed director has stated, "If we can build and support a research infra- structure that seamlessly integrates innovative science with routine clinical care, this will not only help us deliver high-quality results for people in a less resource-intensive way, but will allow us to get into diverse communities where we otherwise haven't been" [1]. PACER aims to support this effort.

PACER leverages recent advances in technology called clinical decision support (CDS) hooks [2] to enable researchers to create and deliver to clinicians very short (under two minutes) research prompts—such as asking a question or doing a brief exam—into the clinical encounter. In doing so, PACER seeks to enable high-speed, real-time, scalable, minimal-risk research through rapid bi-directional transmission of information between a central hub and many distributed electronic health records (EHRs). Utilizing EHRs, our approach operationalizes digital health data exchange standards, such as the Fast Healthcare Interoperability Resource (FHIR), which defines optimal target standards for data elements across healthcare settings, and Substitutable Medical Applications Reusable Technologies (SMART). SMART-on-FHIR apps can pull data from any EHR, patient portal, or data warehouse [3–5]. This distributed approach allows research prompts to be sent to any participating clinician across the country, regardless of their location or which EHR they use. The responses to the prompts (e.g., answers to questions or results of the medical exam) would be documented in the medical record, thereby providing data about a symptom not typically asked about or physical finding not regularly included in a clinic visit. This additional information could also be returned to the central hub to rapidly answer research questions. The following example about how PACER might have been useful during the recent COVID-19 pandemic further explains PACER and its potential impact. After clinicians noted that some patients with COVID-19 experienced anosmia, PACER could have sent prompts to clinicians around the country whose patients had symptoms suggestive of COVID-19 requesting that the clinician inquire about anosmia. This information could have been paired with COVID-19 test results to rapidly confirm or deny an association between anosmia and COVID-19.

We believe PACER could improve the conduct of minimal-risk research in several ways. As a scalable, broad-reaching technology, PACER could accelerate the pace and timeliness of research by quickly fetching data from large numbers of participants. PACER should also increase the diversity of research participants by extending recruitment efforts into more clinical settings, particularly those outside of academic medical centers (AMCs) [6–11]. Because most research occurs at AMCs but most clinical care does not, PACER could mitigate the current, problematic siloing of clinical research. Further, PACER does not require additional research or technical staff, making conducting research more cost-efficient [12]. Thus, more widespread uptake of research activities would be possible with increased clinic participation, improved subject diversity, and the removal of personnel cost-barriers.

While PACER offers the potential to accelerate the pace of research, improve generalizability, and expand diversity among research participants, this approach also presents ethical, sociotechnical, and implementation challenges. Possible concerns relate to the impact of PACER on clinic workflow, the patient-clinician relationship, how consent is obtained, whether compensation is offered and to whom, how privacy and confidentiality are maintained, and whether and how study results should be provided to participants [12].

The purpose of the current study was to obtain insights from patients, clinicians, research experts (individuals involved in developing and conducting research), and bioethicists on potential benefits and challenges presented by PACER. By exploring and describing diverse viewpoints related to PACER, and by identifying potential challenges imposed by PACER *a priori*, we intend for this work to inform the creation of guidelines for future PACER implementation.

## Materials and methods

We conducted one-on-one, qualitative semi-structured interviews with patients, clinicians, research experts, and bioethicists between February 1 2022 and August 31 2022. Verbal consent to participate in this study and for the interviews to be audio recorded was obtained by the interviewer prior to starting the recording of interviews. This study was determined to be exempt by the Northwestern University and the Ann & Robert H. Lurie Children's Hospital Institutional Review Boards (NU IRB STU00215847 and Lurie IRB 2021-4774). No funders had a role in the study design, data collection and analysis, decision to publish, or preparation of the manuscript.

### Interview guide

We created a semi-structured interview guide to explore the integration of PACER into clinical practice. The interview guide, which was iteratively developed to probe possible ethical, sociotechnical, and implementation challenges presented by PACER as identified by the study team, was pilot-tested with three individuals (one patient and two clinician-researchers). To explicate details of the various topics, we chose the basic question "Is the patient having knee pain?" as an example of the added question PACER would embed into a clinic visit. The final interview guide addressed the following topics 1) General reactions (i.e., pros/cons); 2) Use of clinic time; 3) Impact on patient care; 4) Impact on the patient-clinician relationship; 5) Compensation; 6) Patient privacy; 7) Consent; 8) Communicating study results; and 9) Participants' views on two additional specific use cases: a) Questions that probe sensitive topics (e.g., issues about sexual health), and b) Including a physical examination (e.g., assessment for knee crepitus). Interview guide questions were open ended with one exception. The closed ended question asked participants to choose one of the following strategies for obtaining consent for PACER: 1) Waiver of consent (i.e., if the host institution determines that no formal consent from patients is needed, study questions could be asked as part of a visit); 2) Oral consent alone (i.e., patients would be provided a brief explanation of the study question or exam during the appointment and would be offered an opportunity to consent orally); 3) Oral consent plus written information (i.e., patients would be provided a brief oral explanation of the study or exam and would be offered an opportunity to consent orally. After patient questions were answered, patients would receive written information about the study as part of their visit paperwork summary); or 4) Signed consent form (i.e., patients would be provided a consent form explaining the study, either before or during the clinic visit, and would be offered an opportunity to review and sign electronically or on paper if they chose to participate). The interview guide also included questions about participant demographics. (See S1 for Interview guide).

### Participant recruitment and data collection

Candidates were eligible to participate if they spoke and read English fluently, were > 17 years of age, and described themselves as one of the following: clinic staff (including clinicians and administrative staff), bioethicists (people involved in an academic bioethics program

in the Chicagoland area), research experts (principal investigators or study team members), or patients. We used a multi-pronged, purposive snowball recruitment approach. We recruited bioethicists by sharing information about the study with a group of local bioethicists across Illinois. To recruit clinicians and staff, we provided information about the study to the Northwestern Network of federally qualified health centers and private practices via emails in partnership with Feinberg School of Medicine's research network and the Chicago Health Information Technology Regional Extension Center (CHITREC) [13]. For researcher recruitment we contacted some known researchers directly via email. Additionally, some respondents from the outreach to the CHITREC were researchers. We recruited patients via the ResearchMatch database (a national health volunteer registry created by several academic institutions and supported by the U.S. National Institutes of Health as part of the Clinical Translational Science Award [CTSA] program) [14]. We limited patient participation to those in local Chicago-area zip codes. We requested participation from 60 patients identified using ResearchMatch. Prior to confirming their involvement in the study, potential interviewees received an information sheet, sent electronically via email. The information sheet included a brief description of PACER, the study's aims, the risks and benefits of participation, and study and institutional contact information. When the interviews commenced, the interviewer gave a brief description of PACER and how PACER would function and offered the participants the opportunity to ask their own questions prior to the interview questions. All interviews were conducted by one study team member (EF) over Zoom (Zoom Video Communications Inc., San Jose, CA). Audio was recorded and professionally transcribed. Transcriptions were then reviewed for accuracy and de-identification.

## Data analysis

We analyzed the transcript data using domain analysis, combining deductive and inductive components [15,16]. We created topic codes, reflecting the predefined interview guide topics, and input from two team members (KNM, EF). We assessed our team's coding reliability across various topic codes using a subset of the data and Kappa scoring; the Kappa scores for group members ranged from 0.74 to 0.89. Two team members (KNM, EF) compiled text based on topic codes for all transcripts. Data in each topic code was then inductively analyzed by having two team members create and agree upon subcodes for each topic in an iterative manner. When a coding dictionary was agreed upon, two team members separately applied the subcodes for each topic area and resolved discrepancies by consensus. Using Excel version 16.82 (Microsoft, Redmond, WA), a Zipf analysis was conducted to review the codes created in the content areas. Use of Zipf analysis is based on Zipf's Law, which holds that in natural language utterances, the frequency of any word or concept is inversely proportional to its rank in the frequency table [17]. Thus, common concepts should appear in the first few transcripts, and each subsequent transcript analyzed should add fewer and fewer new concepts. Because this phenomenon follows a power law distribution, it is possible to fit this curve and calculate the theoretical number of concepts in a corpus or domain without having to analyze an infinite number of transcripts. Our Zipf analysis showed exceptional fit with $R2 = 0.9714$ for all codes and $R2 = 0.9786$ for the subcodes within topic codes, demonstrating saturation (see S1 Fig).

The full study team then reviewed the data and reached consensus on key domains reflecting the overlapping nature of subcodes within the different topic areas. We then identified themes and subthemes within each domain. Because the overall study was not designed to represent information quantitatively, we describe our results qualitatively. We do, however, use qualitative descriptors (e.g., "few," and "most") to give readers some sense of how frequently a particular idea was expressed. Qualitative analysis was done using Dedoose version

9.0.17 (SocioCultural Research Consultants LLC, 2021) [18]. Reported race was categorized into National Institutes of Health categories and ethnicity into categories developed by two team members (KNM, SS) through an iterative process of reviewing and combining participant responses. We calculated descriptive statistics for responses to the one question asking about consent and demographic data using Excel (version 16.82).

## Results

We interviewed 47 people. See Table 1 for participant demographics. We identified the following domains in our interview transcript analysis: 1) Impact on clinical research; 2) Consent; 3) Compensation; 4) Impacts on people and organizations; and 5) Implementation. Below we describe the themes and subthemes associated with each domain. Table 2 provides definitions of the domains, themes, and subthemes, as well as exemplar quotes.

**Table 1. Demographics of study participants.**

| Demographic variable | N (%) |
|---|---|
| Gender | |
| Female | 31 (66) |
| Male | 15 (32) |
| Non-binary | 1 (2) |
| Role | |
| Patient | 21 (45) |
| Research experts/bioethicists[a] | 12 (26) |
| Clinicians/clinic staff[b] | 14 (30) |
| Self-identified race[c] | |
| Asian | 5 (11) |
| Black | 8 (17) |
| White | 29 (62) |
| Not disclosed | 4 (9) |
| Mixed | 1 (2) |
| Self-identified ethnicity | |
| American | 1 (2) |
| East Asian | 2 (4) |
| European | 5 (11) |
| Hispanic | 6 (13) |
| Jewish | 4 (9) |
| Mixed | 3 (6) |
| Non-Hispanic | 4 (9) |
| Not disclosed | 16 (34) |
| White | 6 (13) |
| Age in years, mean (median, range) | 46 (45, 25 – 77) |

[a]This group includes 8 people who identified themselves as bioethicists (of those, 4 noted specific expertise in research ethics, 2 were lawyers, 3 educators, 3 researchers, and 1 IRB member) and 6 identified themselves as research experts (of those, 3 noted they were bioethicists, 2 research coordinators, and 4 researchers). Some people described themselves as both bioethicists and research experts.

[b]This group included 8 clinicians (7 physicians and 1 nurse) and 6 clinic staff (2 people involved in compliance/quality improvement, 2 practice managers, 1 medical assistant, and 1 pharmacist).

[c]Based on the National Institutes of Health Race Categories.

**Table 2. Domains, themes, and subthemes with exemplar quotations[a].**

| |
|---|
| **Domain 1 Impact on clinical research: Views on the potential value of PACER for conducting clinical research** |
| *Theme 1 Research participation:* PACER's impact on patient or community participation in research |
| "I think that academic medical centers are elite spaces. …So having it in more in satellite clinics or like in, a more like in different types of communities and expanded, I think that reaches a wider population, which means that we have more data and more information and more diversity to understand population health." ID 50 (Patient) "You're kind of capturing a very large group of people. …you're probably capturing a very diverse group of people. So, whether that's like age, gender, sexual orientation, you know, race all of that stuff. …the other nice part about it too is like okay well the patient doesn't need to go anywhere, you know what I mean? They're just having it done in their doctor's office" ID 25 (Clinic Staff) |
| *Theme 2 Research findings:* Impact of PACER on the quality of research findings |
| "Well, I think the positives are getting much better at data for the future, and everyone would be helped by that." ID 57 (Patient) |
| **Domain 2 Consent: Views on the potential impact of PACER on obtaining consent to participate in research** |
| *Theme 1 Rationale for consent types:* Explanation for the consent type |
| *Subtheme 1a Study factors:* How factors related to the research study impact consent type |
| "Yes, for cases like knee example given, I think with oral consent is perfect cause that's minimal risk, yes. But if the focus is like HIV, cancer—I think both the oral consent and written should cover that." ID 56 (Patient) |
| *Subtheme 1b Clinic factors:* How clinic characteristics and issues impact consent type |
| "A signed consent form adds time, complexity, interruption...if the written information could be passed off like in the MA check right, and just say, 'Hey, we're doing a little—we're doing some research on a medication you're taking. Here's just some information, right if you're comfortable, the provider may ask some questions, and you don't have to say, if you want to, and here's a little bit of information you can read while you wait for the provider to come into the room' that's pretty fluid." ID 23 (Clinic Staff) |
| *Subtheme 1c Trust:* How the type of consent could impact trust between patients and clinicians |
| "And we need to give them the freedom to exclude themselves because you don't want to do research to the point where it damages communities' trust." ID 50 (Patient) |
| *Subtheme 1d Signed document:* Pros and cons to using a signed documents as part of the consent process |
| "I think, the oral consent plus the written information. And mainly because the signed consent form in my experience, typically actually don't lead to better informed consent." (ID 31) (Research/Bioethics Expert) "I think the signed consent form sent before so the patient can refer to it. And it's a hard copy would be preferable, and that way it doesn't take that much time from the clinical encounter." ID 18 (Clinic Staff) |
| *Subtheme 1e Written information:* Impact of including written information as part of consent process |
| "Some people process things better orally. Some people are better visual learners. So giving them that combination gives you the best coverage, you know, and is supportive a wide range abilities." ID 51 (Patient) |
| *Theme 2 Scope of consent:* Input about a "uniform" consent process (using one consent for multiple studies) |
| *Subtheme 2a Permissibility of "uniform" consent:* Comments in support of "uniform" consent |
| "…they have a basic consent form that's already done and already signed. And then if there's anything additional for the studies, then they could look over and sign whatever additional." ID 47 (Patient) |
| *Subtheme 2c Dissent of "uniform" consent:* Comments in dissent of "uniform" consent |
| "I personally think there should be a new one each time just so the patient knows and can like consent or like revoke consent on their own basis." ID 59 (Patient) |
| *Theme 3 Consent process:* Comments about the process of obtaining consent |
| *Subtheme 3a Information:* Importance of information comprehension |
| "I would, I would say oral with the written gives them some information to go along with it, so they'll understand what it is." ID 58 (Patient) "How do we implement meaningful informed consent around this? And I'm flagging that I think the current inform consent processes are very flawed." ID 15 (Clinic Staff) |
| *Subtheme 3b Timing:* Timing considerations related to the consent process |
| "I think it would be helpful if that was done ahead of time" ID 57 (Patient) |
| *Subtheme 3c Recruitment personnel:* Comments about who should be involved in the consent process |
| "This is something that the receptionist could handle in the beginning, especially." ID 27 (Clinic Staff) |
| **Domain 3 Compensation: Views on the potential impact of PACER on compensation** |
| *Theme 1 Clinic/clinician compensation:* Compensation to the clinician and/or clinic for research involvement |
| *Subtheme 1a Rationale for compensating:* Support for compensating clinics and/or clinicians |
| "…research is work and folks are conducting research and should be compensated for that work." ID 13 (Research/Bioethics Expert) |
| *Subtheme 1b Rationale for not compensating:* Support for not compensating clinics and/or clinicians |
| "I find that incredibly troubling, because now they have financial conflict of interest. …there's tons of research on that like that's how humans work." ID 29 (Research/Bioethics Expert) |
| *Subtheme 1c Patient awareness:* Importance of patient awareness of clinic/clinician compensation |
| "…but certainly the level of compensation for the doctor, the patient, the clinic, and transparency around that would all be very important." ID 15 (Clinic Staff) |

*(Continued)*

**Table 2.** (Continued)

| | |
|---|---|
| ***Theme 2 Patient compensation:*** Compensation to patients for research involvement | |

*Subtheme 2a Rationale for compensating*: Support for compensating patients

"…some kind of payment that shows 'Yes, we're asking for your time'" ID 3 (Research/Bioethics Expert)
"…if companies are going to collect my data, I want to have a part in the profit they make off of selling my data." ID 59 (Patient)
"I they could [compensate patients], I think, if it helps increase participation." ID 24 (Clinic Staff)

*Subtheme 2b Rationale for not compensating patients*: Support for not compensating patients

"I think paying people for research is, is a bad idea. And that gets into the paying people for body parts, paying people for eggs, you know, it makes it quite sordid." ID 12 (Research/Bioethics Expert)

*Subtheme 2c Types of patient compensation*: How to compensate patients

"…I think compensation for any additional travel or services they'd need to partake in." ID 44 (Clinic Staff)
"I think a, you know, a $5 Amazon gift card seems to be fair compensation for people's time." ID 50 (Patient)

**Domain 4 Impacts on people and organizations:** View on the potential impact of PACER on people (patients, clinicians, and clinic staff) and organizations

***Theme 1 Clinic:*** Impact of PACER on clinics and clinic staff

*Subtheme 1a Financial burden*: The potential financial impacts of PACER to clinic operations

"Because time is money in the practice." ID 19 (Clinic Staff)

*Subtheme 1b Space*: The office space impacts of PACER to clinic operations

"I think the idea of taking the patient to another area would be best, because then it would like free up your exam room, too, so you can keep kind of going on schedule." ID 25 (Clinic Staff)

*Subtheme 1c Staff*: The impact of PACER on clinic staff

"The ideal thing would be to have this kind of questions put in for the medical assistants who are rooming the patients…they could ask this question" ID 20 (Clinic Staff)

*Subtheme 1d Time burden*: The impact of PACER on timing of clinic activities.

"…if they're spending a lot of time asking questions or spending more than a minute or two, I would think that…it might affect other people." ID 57 (Patient)

***Theme 2 Patient:*** Impact of PACER on patients

*Subtheme 2a Patient care:* How PACER would impact patient care

"I guess find out more about different medications, side effects, combinations of medications. For that personal regimen that patient might benefit from. I think that'd be positive." ID 18 (Clinic Staff)

*Subtheme 2b Time burden:* The impact of PACER on time-related burdens to patients

"But if I've got two minutes with my doctor and a minute or 30 seconds is going, dealing with an external issue, that's problematic. I think." ID 51 (Patient)
"I'm literally really paying you to be here, and now you're gonna take more of my time than was necessary, you know." ID 29 (Research/Bioethics Expert)

*Subtheme 2c Privacy:* Issues related to patient privacy

"So it doesn't matter to me whether they can access that or not cause you can see my entire medical history on MyChart. So, I, this is just an extension of that, like it doesn't change anything." ID 50 (Patient)
"But I really don't think patients want their names out there, their date of births out there none of that. As long as the data is de-identified, I think it will be great." ID 34 (Clinic Staff)

*Subtheme 2d Financial burden:* The potential financial impact of PACER on patients.

"And also like, especially if you pay for appointments out of pocket, it kind of feels like, 'Wait, why are we focusing on this?' You know, 'When I'm paying you to, for this issue,' right?" ID 59 (Patient)

***Theme 3 Clinician:*** Impact of PACER on clinicians

*Subtheme 3a Patient care:* The impact of PACER on clinicians' ability to provide patient care

"I think the more that doctors are involved in outside of just their general practice…it just keeps everybody like more educated and more up to date, with everything." ID 22 (Research/Bioethics Expert)
"…the clinician an opportunity to improve their, their knowledge of what the patient is experiencing and provide additional care?" ID 52 (Patient)

*Subtheme 3b Additional responsibilities:* The impact of PACER on clinician responsibilities

"…it could be like distracting…any added things to do when you're already in a like high-stress environment where like might make mistakes and all that, um, could be like challenging for sure." ID 28 (Research/Bioethics Expert)
"…there are a number of pop-ups and I don't mind when they're relevant, but I think the thing that can be really disruptive to clinic flow is when they require a response." ID 44 (Clinic Staff)

*Subtheme 3c Time burden:* The potential time burdens associated with PACER

"…if I have 15 minutes with someone and we weren't planning an exam already, that would be more of a burdensome ask to add one in." ID 44 (Clinic Staff)

*Subtheme 3d Financial burden:* The potential financial impact of PACER on clinicians

"…it's a question, whether that impacts a physicians' earnings or salary. …they would be more inclined to do these research activities because they're not losing income in patients." ID 50 (Patient)

*(Continued)*

**Table 2.** (Continued)

| |
|---|
| *Theme 4 Clinician-patient relationship:* Impact of PACER on the clinician-patient relationship |
| *Subtheme 4a Disruptive:* How PACER could disrupt the clinician-patient relationship |
| "'Oh, well, the doctor is doing research now, so maybe there's some financial incentive for them to be doing this and like, can we trust them?'" ID 25 (Clinic Staff) |
| *Subtheme 4b Supportive:* How PACER could support or enhance the clinician-patient relationship |
| "It expands their relationship, and the patient feels more cared for in the process of them asking other things so." ID 60 (Patient)<br>"I think that can help the patient physician relationship grow and develop further into always trusting the physician's always looking out for the well-being." ID 18 (Clinic Staff) |
| *Subtheme 4c Neutral:* How PACER would not significantly impact the clinician-patient relationship |
| "I think it potentially could [impact the clinician-patient relationship], but I don't think it has to." ID 31 (Research/Bioethics Expert) |
| *Subtheme 4d Clinician role duality*: How PACER could create a dual role for clinicians |
| "There's the respect of a reciprocal relationship that you were my patient, and I was exclusively focused on your needs for the last 20 minutes. Now I'm about to switch roles and put a researcher hat on, and now you're gonna we're gonna focus on my needs which are to think about your knee." ID 29 (Research/Bioethics Expert) |
| **Domain 5 Implementation:** Views on PACER implementation |
| **Theme 1 General implementation considerations:** Views on general PACER implementation |
| *Subtheme 1a Ensuring patient understanding* |
| "It's important to have that community education so that whoever is asked understands …that everyone is going to be in this situation. …that goes a long way to build trust." ID 16 (Research/Bioethics Expert)<br>"We have a lot of people who don't speak English. Um, so as long as they're like thoroughly having an understanding of what they're consenting to then that's great." ID 47 (Patient)<br>"How will patients know what's routine care versus non-routine care?…And if people don't know that that's a problem." ID 32 (Research/Bioethics Expert) |
| *Subtheme 1b Patient-centered design suggestions* |
| "…instead of verbally with the primary care physician, they could possibly be handed a tablet with the questions on there." ID 53 (Patient)<br>"I mean, if it's done solely through MyChart, I think that the worry is that older patients or patients who aren't technologically savvy might not access that." ID 50 (Patient) |
| *Subtheme 1c Clinician-centered design issues* |
| "…a pop-up should never be the first, only, or best way of alerting clinician to do anything …where in the workflow that's done actually needs to be thought about very, very, very carefully." ID 35 (Clinic Staff)<br>"I think it's fine as long as it's brief and to the point." ID 27 (Clinic Staff) |
| *Subtheme 1d Question frequency/quantity* |
| "…it depends on an individual doctor, and how he manages his workload and workflow, and how efficient it is. Whether, you know, it could be minimal additional work, and time." ID 27 (Clinic Staff)<br>"They should just explain that to the patient ahead of time, "I have five questions. I have eight questions. It's gonna take this amount of time," and then see if the patient is willing to cooperate, so." ID 60 (Patient) |
| **Theme 2 Post-study contact:** Views specific to contact with patients after the clinic visit in which PACER-type research was included |
| *Subtheme 2a Need for additional care:* Addressing patient care needs surfaced because of involvement in PACER |
| "…if it reminded them that they have knee pain, chronic knee pain, and we can help them good for both parties." ID 19 (Clinic Staff)<br>"In in the scenario you're describing [the physician is] in a dual role. …Even if you're a researcher, right, if the patient brings something up significant, or you know, or even minor right, you have that professional obligation to say, you know, "You need to see someone about that." ID 16 (Research/Bioethics Expert) |
| *Subtheme 2b Follow-up questions:* Views about follow-up questions after contact with PACER in the clinic |
| "I think they should go through the medical practice or the doctor, the nurse, so that they could send them an email, an automated email saying, you know, 'This study is asking some other questions, is it okay if, if they contact you and what's the best way to do it.'" ID 57 (Patient) |
| *Subtheme 2c Sharing study results:* Views about sharing study results |
| "Yes, because the doctor should be able to learn from this." ID 52 (Patient)<br>"Those disseminations may look different …for a professional audience as compared to a community audience." ID 23 (Clinic Staff) |
| **Theme 3 Unique situations:** Views about using PACER to add an exam (versus a question) or a question about a sensitive topic to the clinical encounter |
| *Subtheme 3a Exam versus question:* Views about using PACER to add an exam to the clinical encounter |
| "…when it's just the knee and like not invasive, as long as you have consent from the person, I don't see any issues." ID 40 (Patient)<br>"…they're actually adding additional steps in the examination process…they should be compensated." ID 58 (Patient)<br>"So, if there's a compelling reason why examining the knee is helpful, if somebody tells you their knee hurts, you're gonna look at their knee, so that flows. If it is a more intimate exam, or a more invasive exam, then I would start to raise more questions about it." ID 23 (Clinic Staff) |

*(Continued)*

**Table 2.** (Continued)

*Subtheme 3b Sensitive topics:* Views about using PACER to add a question about a sensitive topic to the clinical encounter

"And because that type of question could lead to some discussions that the first screener person may not be fully qualified to answer. Yeah, that's a good point." ID 31 (Research/Bioethics Expert)

"I think there would have to be a more stronger emphasis on the de-identification. For, like certain scenarios like HIV or of venereal diseases. And with also having that informed consent component, I think the patient can decide to do it or not depending on their level of sensitivity, to the issue." ID 18 (Clinic Staff)

[a]Minor text edits to quotes were done to improve context and flow.

## Impact on clinical research

Most participants indicated that PACER could improve research participation and advance scientific knowledge. Particularly, patients noted that PACER may improve how clinical research is conducted. Some stressed that PACER could create a unique opportunity to connect clinical research with patient communities. A few participants commented that PACER would be a good use of technology.

## Consent:

Fig 1 shows responses to the question about what type of consent is appropriate. The majority (36%) of participants favored obtaining oral consent combined with providing written information. Only one person favored a waiver of consent. Figs S2 and S3 show responses to the question about the type of consent by participant role and gender, respectively. We describe three themes related to consent below: Rationale for Consent Type, Scope of Consent, and Consent Process.

**Consent type.** In explaining their preferred consent type, participants described factors related to the study, the clinic, the impact on trust, having to sign a document, and providing written information. In considering study-specific factors, they commented on study risks, with some participants suggesting that the consent process should be adapted based on the research question, e.g., a question on a more sensitive topic or research involving a physical exam might warrant a different type of consent compared to research

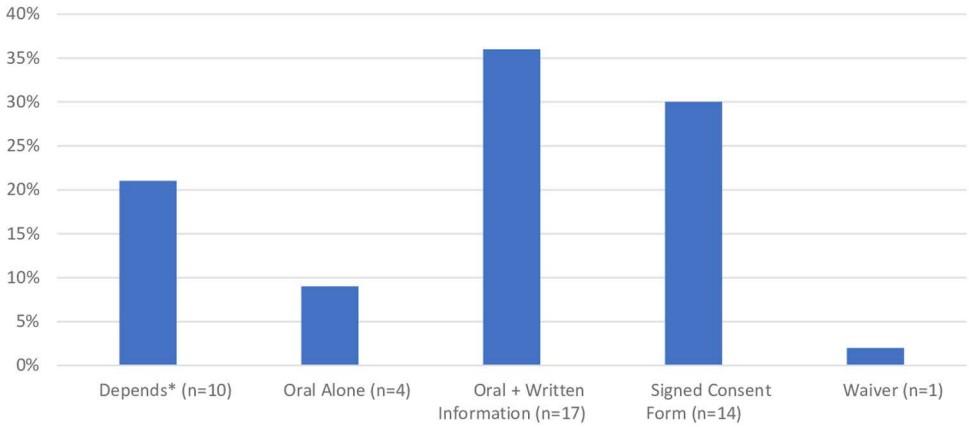

**Fig 1. Choice of consent type based on percentage of all participants (n = 47).** Asterisk indicates responses coded as "depends" and anyone who endorsed more than one response. One person did not endorse any of the consent types presented.

involving asking a non-sensitive question. A few noted the importance of the clinical setting on the type of consent sought, including efficiency, interruption of workflow, and the importance of ensuring that the consent process would not hinder patient clinical needs during a visit. Several participants noted that the consent type could impact trust between clinician and patient; therefore, fostering trust and mitigating mistrust should be prioritized when choosing the appropriate consent type. One participant claimed that, depending on how consent was obtained, there is a risk that some patients might feel coerced into participating.

Participants also discussed specific aspects of the documentation associated with the consent process. Several participants noted the pros and cons of having a signed consent form, with nearly equal representation of both views. People who felt a signed consent form was not needed cited a variety of concerns. These concerns included: having to sign something might deter patients from participating, signing might create challenges associated with organizing and managing the documents, signed consents do not lead to better informed consent, and requiring a signature adds time to and disrupts the clinic visit. Participants who supported signed consent noted the benefit of having a proof of study participation agreement; protection from potential future liability; adherence to research norms; and the benefit to participants of having a document with information about study details, contact information, etc. Participants explained that having a written document (e.g., flyer) was valuable, even if it was not part of a signed consent form. Participants described how such a document could remind patients of the details to which they consented and could be used as a reference regarding their participation in research, both during and following their clinic visit. The need for clear comprehension of research goals and expectations from participants was reiterated by many participants. One participant noted that patients may differ in their comprehension of written versus oral information.

**Scope of consent.** Most participants did not support having a "uniform" consent process (in whatever form that might take) that applies to multiple PACER studies wherein a participant's single, one-time consent would be reused for consent to participate in multiple research studies. Most participants stressed that studies involving asking a new question or performing an examination would warrant obtaining a new consent tailored to the specific study. However, a few participants commented that having a uniform consent would be permissible when studies address similar issues. A few participants felt that uniform consent was reasonable with some caveats, such as mandating renewing the consent after a certain time period; clarifying the process so that patients understand that their consent applies to multiple studies; or obtaining a generic consent that indicates the patients' willingness to participate in these types of studies, but also requires that patients are provided specific information about each new study and are allowed to opt-out if desired.

**Consent proces.zs.** Participants commented on three elements of the consent process: the information provided to patients, the timing, and the actors involved. Most participants commented that information should be presented to patients in a comprehensible fashion to ensure understanding of the research process. A few stressed the importance of ensuring understanding because current approaches to obtaining informed consent do not always lead to informed participants. Others expressed that the consent process should clarify the clinician's dual role as both clinician and researcher, as well as make explicit details related to compensation or potential conflicts of interest. Some participants highlighted that informing local communities about a particular clinic's involvement in PACER-style research could promote trust (e.g., by addressing misconceptions or concerns that certain communities are being singled out or exploited).

The timing of the consent process was explored by several participants. Some suggested that consent should occur during the clinic visit, but others felt consent should commence prior to the clinic visit. Pre-visit consent procedures could minimize the impact on clinic workflow, and any community-directed research education could happen prior to clinic visits. Another key facet of consent timing mentioned by participants was the importance of ensuring that sufficient time is allocated to the consent process. This is necessary to ensure that patients fully understand what is being asked of them and to avoid coercion. However, other participants cautioned that too much time dedicated to consent could impact the time available for patient care.

Finally, participants discussed who should obtain consent from patients. A few participants noted that a clinic administrator could disseminate written information to potential study participants. Some mentioned that involving a nurse or medical assistant could reduce the burden on the clinician whom the patient came to see (e.g., physician or advanced practice provider). A few participants suggested that the physician introduce the study, given that patients might feel more comfortable if their physician endorsed the research. Alternatively, one person noted that having someone other than the physician both introduce the research and ask the research question could help to differentiate the research part from the clinical part of the visit.

## Compensation

Participants commented on compensation provided to the clinic and clinicians, as well as patients, for their involvement in research. The data presented focuses on the provision of financial compensation, because questions related to non-financial compensation were not asked consistently across interviews.

**Clinic/clinician compensation.** Most participants suggested that the clinic and/or clinicians should receive some kind of compensation for involvement in PACER research. Those in favor of such remuneration stressed concerns about the additional work, time required, and the potential impact on the number of clinic patients seen. One participant claimed that providing clinic and/or clinician compensation showed "goodwill." Another participant noted that patients should be made aware if clinicians and/or clinics received compensation. Several participants who supported compensation to clinicians and/or clinics also described reasons to withhold such compensation. Some observed that, if the clinician or clinic received compensation, there should be assurances that such compensation would not impact the integrity of the providers and patient care, which could be undermined by shifting incentives (e.g., if incentives were tied to the number of participants enrolled). Participants described concerns that compensation could influence how patients would be treated. Others felt that merely asking an additional question during a visit would not warrant compensation. One person noted that physicians have an obligation to give back to their community and, therefore, should not receive compensation.

**Patient compensation.** Most participants felt that patients should receive some kind of compensation for participation in PACER research. Advocates of patient compensation highlighted that compensation incentivizes participation, enhances participant diversity, acknowledges the additional time required of patients, and supports travel or other expenses required for study participation. Participants also noted that compensation is warranted because the research involves collecting personal data, researchers could potentially profit from the results, and PACER may ask patients to do something that is unrelated to the reason for their clinic visit and may not impact their health. Several participants felt that patient compensation was not necessary if participation involved minimal time or only a few questions. Some claimed compensation would not be necessary because involvement in PACER might educate participants about the research topic and advance science.

## Impacts on people and organizations

We discuss the potential impacts of PACER on people (clinicians, staff, and patients) and organizations (clinics) together, as many subthemes overlapped.

Two subthemes identified were time and financial burdens. Most participants noted that PACER could increase the time needed for patient visits, potentially worsening clinic work-flow and patient throughput. Because of the additional time required for PACER, clinicians might see fewer patients, and patients might experience longer visits or give up precious time with their clinicians for issues not related to their health. For both the clinic and clinicians, decreasing the number of patients seen could have negative financial impacts. This financial burden might be assumed by the clinic or clinicians, or it could be passed along to patients by increasing the cost of their visit or creating waiting lists.

Another subtheme related to how PACER could improve patient care. Several participants noted that PACER might encourage physicians to ask more questions and address more patient concerns, thereby increasing their knowledge of the patient. Some of these participants felt that physician participation in PACER research would evince a willingness by the clinicians to dedicate themselves to improved patient care. Moreover, some participants noted that PACER could advance medical knowledge (generally and for individual clinicians) and might support better resource allocation.

The issue of privacy was a key topic. Accessing patient data that has been de-identified would be reasonable, according to several participants. A few commented that following HIPAA guidelines was important to ensure privacy and that data should be stewarded by research teams. Nevertheless, other participants were not particularly concerned about privacy. Some described how patient data was already accessible, available, and being used by third parties; thus, protecting privacy was no longer possible. Several participants noted that transparency, i.e., awareness by patients about how their data were used and who had access to their personal information, was important.

Another issue discussed was how PACER might add to clinicians' responsibilities and burdens. In addition to adding another task to the clinic visit, PACER would require that clinicians receive some specialized education or training. This instruction would involve informing clinicians about what question to ask or the type of exam to perform, as well as what actions to take if a patient concern is uncovered or an examination reveals an abnormality. Also, some noted disruptions to clinicians' workflow and challenges associated with attending to EHR research alert "pop-ups." One physician participant explained that clinicians are already burdened by obligatory documentation related to patient care and opined that many might be wary of additional demands.

Some participants expressed concerns about using limited clinic space (e.g., rooms) for PACER activities, such as obtaining consent or doing an exam. Others suggested that these activities could be performed by a trained nurse or medical assistant at the time of patient intake while obtaining vitals or history or during the medication management process. Additionally, a few noted that PACER could add to the activities of clinic staff (e.g., consenting or informing patients about PACER or asking the research questions) and could blur staff roles, making it unclear when staff are acting in service to the patient or to the research.

Most participants addressed the impact of PACER on the clinician-patient relationship. Several participants noted that PACER could improve this important relationship. By the clinicians asking more questions during a visit, patients might perceive greater concern for their well-being. Several participants felt that seeing their clinician participating in research and demonstrating a desire for ongoing learning could build trust. One participant claimed that offering patients the option of participating in research could improve the clinician-patient relationship.

However, several participants expressed concern that PACER could disrupt the clinician-patient relationship. Some worried that PACER could fuel distrust, particularly for those unfamiliar with research or for those who might perceive their involvement in research as part of an experiment. A few remarked that mistrust could also occur if patients felt clinicians were involving them in PACER for financial gain. Others observed that patients might perceive their clinician as underqualified if they asked odd questions unrelated to the focus of the clinical visit. Finally, several participants discussed clinicians taking on a dual role, as clinician and as researcher, noting that blurring these lines could erode trust. Thus, many participants suggested that efforts should be made to ensure that patients are aware of which parts of their visit focus specifically on patient care and which are focused on research. Conversely, some participants did not think PACER would impact the patient-clinician relationship, particularly with proper consent and patient education about the research.

## Implementation

**General considerations.** Regarding PACER implementation, people commented on ensuring patient understanding, described opportunities to increase patient- and clinician-centeredness, and commented on the number and frequency of research questions. As previously noted, a few participants felt that providing education to the community could improve understanding about the research. Some recommended that the clinician explicitly communicate that the question is being asked solely for research purposes and is not part of standard care. For pragmatic considerations, a few participants suggested that the research question come at the end of the visit, contingent on available time. Relatedly, a few people asserted that the Clinical Decision Support prompts should be brief. When asked about question frequency and quantity, participants discussed both clinician- and patient-level considerations. A few participants felt that the number of questions could vary by clinician or clinic depending on the practice organization, number of patients seen, and even clinician personal style. For patients, a few people suggested that, if multiple questions were included in a visit, patients should be informed about the number of questions and time involved. Caution should be taken to ensure that the number of questions does not take so much time from the clinical encounter that it negatively impacts the patient's care. Also, asking multiple questions of one patient should prompt consideration of additional patient compensation. To increase patient understanding of the questions, some noted the importance of having materials suitable for non-English speakers.

**Post-study contact.** Discussions about contact with patients after the clinic visit in which PACER-type research occurs included addressing additional care needs revealed as a result of the research, permissibility of follow-up questions prompted by the initial research question, and sharing of study results. Nearly all participants commented that, if involvement in the research revealed a new health issue, the clinician should take the appropriate steps to address it during the visit, at a subsequent visit, or by making an appropriate referral. Several people felt that identifying a new problem is a potential benefit of this research approach. With PACER, there is the possibility that results from the initial question or exam could prompt a researcher to want to reach out to patients with additional questions. Several people noted the reasonableness of contact after the clinic visit to ask an additional research question but cautioned that information about this possibility must be included in the informed consent process. A few people advised against asking patients additional questions, claiming that doing so would complicate the consent process, potentially involve a third party (if these questions aren't asked by the clinician), and add an additional patient burden. One person noted that if a patient were asked a subsequent question it should come from the clinician, rather than a research team member who is not part of the clinic and is unknown to the patient.

A majority of people noted that study results should be shared with both clinicians and patient participants. Sharing research results was described by one person as a way of moving research into the community, "to democratize science." Sharing results with clinicians was described as beneficial because it demonstrates that clinicians were partners in the research and doing so provides a learning opportunity for clinicians. A few people noted that whether to share results with patients would depend on the study. Some emphasized that such sharing must be done by the study researcher who is not part of the clinic, not the clinician, and in a manner comprehensible to patients. Others, however, thought results would be best delivered to patients by their clinician. A couple of interviewees suggested creating study summary materials designed for layperson comprehension and disseminating them to participants while providing more detailed information to clinicians. Additionally, several participants acknowledged that not everyone will want to know the results and recontacting all participants could prove logistically challenging.

**Exam versus question.** Nearly half of those interviewed felt that including a specific physical exam in the clinic visit, as opposed to merely asking a question, is acceptable. Several people specified that the exam should be fully explained during the consent process. Others suggested that permissibility would depend on the invasiveness of the exam; e.g., a pelvic exam might not be acceptable, but assessing knee movement might be fine. The added time burdens or inconvenience posed by an additional exam was mentioned by several study participants. A few people claimed including an exam would increase the need for or justify compensating participants. Several questioned the idea of including an exam that has no potential benefit to patients. A few people noted that patient acceptance of including an exam would depend on how much they trusted their clinician. Finally, several people thought that clinicians might need training in how to do the exam properly, adding burden to clinicians.

**Sensitive topics.** When asked if it was acceptable to add a question related to a sensitive topic (such as something about a sexually transmitted disease), several participants noted that in such cases the consent process should be explicit about the subject matter and added that the questions should be relevant to the patient's situation and carefully presented. Participants also noted that including a sensitive question might require that the clinician have education or training to ensure appropriate questioning and preparedness to answer patient questions that might arise. Additionally, and somewhat uniquely to questions around sensitive topics, several people commented on the increased importance of issues related to privacy and confidentiality, noting the need for de-identifying information and enhanced data protections. A few people noted that some patients may choose not to participate if research questions are sensitive in nature.

## Discussion

In this study, we obtained insights from patients, clinicians, clinic staff, research experts, and bioethicists on the potential ethical, sociotechnical, and implementation issues raised by PACER. We identified five key domains: impact on clinical research, consent, compensation, impacts on people and organizations, and implementation. Most reactions to PACER were positive. Participants indicated that PACER could improve clinical research processes and increase research participation, as well as advancing scientific knowledge. At the same time, the results demonstrated diverse views about implementing various aspects of PACER methodology. We discovered a diversity of opinions within each group of respondents. Although the responses were not amenable to a formal statistical analysis, we did not identify obvious trends within or among each category of interviewee. Thus, there was no unanimity regarding a single optimal design for a PACER implementation, and future implementers will need to weigh the various opinions. However, the input we received highlighted several considerations for PACER implementation and suggested aspects for future assessment. The interviewees' perceptions are vital for us to identify and further explicate the crucial features of this research

design so that it can eventually be implemented successfully. And by exploring PACER's ethical implications, we hope to share a sound, permissible, efficacious research methodology with the wider research and bioethical communities, as well as with clinicians interested in engaging with research. Below, we discuss the interview feedback in relationship to the traditional bioethical principles of autonomy, justice/fairness, beneficence, and nonmaleficence. We also suggest what a "Pilot PACER" study might look like to further address the input we received and consider how such diverse perspectives could be respected if PACER is implemented. We believe that this study of participants' perceptions of PACER and its ethical implications, as well as how to implement PACER, can help guide the wider research and bioethics communities—including researchers, bioethicists, institutional review boards (IRBs), and others—who may wish to implement PACER.

Considerations of how to obtain consent for PACER, and who should do so, raise issues of patient autonomy. The informed consent process must enable patients or their representatives to make decisions best aligned with their wishes and values, free from coercion [10]. Participants' input on consent revealed significant variability in views about how to obtain consent. Some comments reflected the concern that truncating the consent process (e.g., requiring only oral permission) risks harming the process's integrity and inhibiting patients from making well-informed choices. However, other participants noted that obtaining written informed consent could prolong the clinical encounter—reducing efficiency and potentially affecting participation. Written consent would also require additional training for and availability of clinic personnel to obtain this consent, impacting cost and scalability. Furthermore, the issue of who obtains consent from study participants is vital to understanding the impact of PACER on autonomy. A patient may be more likely to agree to participate in research if the consent is solicited by a trusted physician, such as a family practice or primary care physician, and they be more comfortable asking questions or being honest about any potential reservations when speaking with someone whom they know. However, some patients might feel uncomfortable refusing a request from their regular provider, thus inadvertently pressuring them into participating. These patients may be concerned about how their physician would perceive their refusal and about the impact that might have on their relationship and future care. On the other hand, if the study topic is particularly sensitive, some patients may not wish to disclose such information even to their regular physician. In such cases, their clinician's involvement might preclude them from participating, and another individual should be involved. Who among the clinic is chosen to obtain consent may strongly influence—even impel—patient participation. Thus, there is a tension between respecting participants' autonomy and maximizing the efficiency and scalability advantages that PACER offers.

Additionally, a related query is whether waiving written informed consent for PACER-type research would comply with current regulations and whether enacting such waivers would have an adverse impact on patient autonomy. Does PACER meet the current U.S. requirements for a waiver of written consent? Some might argue affirmatively provided that the research addresses a question that could not practicably be carried out without waiving consent, any question or exam introduced is considered minimal risk, and that by obtaining oral permission and providing written information PACER would not adversely affect the rights of participants [19]. Nevertheless, 30% of participants in this study felt written consent was required. Notably, only 25% of patient participants advocated for written consent. Justifications for signed consent focused largely on ensuring that patients have adequate, comprehensible information about the study. This emphasis on comprehension aligns with research demonstrating limits to research participant understanding of clinical research, though admittedly obtaining written consent has not been shown to overcome gaps in comprehension [20,21]. Importantly, more than 20% of participants felt that the nature of the study

should determine whether to obtain written consent, noting that more sensitive questions and adding a physical exam might necessitate doing so. To have their autonomy truly preserved and respected, participants must be able to comprehend their available options and the risks and benefits of research participation. Given this participant feedback, future work on PACER implementation should consider if waiving written consent and using oral consent alone or in conjunction with the provision of written information can still preserve adequate participant comprehension and respect participants' autonomy.

Another important, ethically-complex component of consent relating to both participant justice and autonomy pertains to compensation. Typically, compensation to research participants is thought to enhance both recruitment and retention. Experts recommend reimbursing participants for their expenses and contributions, while avoiding both under- and overpayment to research participants [22]. Ethical concerns relate to inducement and exploitation [22]; as such, compensation raises issues of justice and fairness. Given that participants must (typically) pay to see their clinicians, and that the visit often incurs additional costs such as time away from work and transportation, remuneration for participation in PACER type research might seem appropriate to some people. Some might view it unfair to take time from the clinic visit for research without some kind of compensation. Yet, it is essential that compensation not lead to coercion. Given the possibility that it could unduly influence decision making, compensation can also impact participant autonomy. Financial or other incentives must not disproportionately influence or pressure the patient into deciding whether or not to participate in research. This is especially true when the patient may not derive a direct medical benefit from study participation. Participants should not have their decision fundamentally altered by the existence or the degree of any form of compensation.

Participants in this study also highlighted the importance of providing compensation to the clinicians and/or clinics. In most research, determining how to appropriately compensate participants for research is enigmatic; adding the complexity of compensation for clinicians and clinics seems to further complicate the matter. Is it possible to calculate the cost of two minutes during a clinic visit? If so, how would such costs be divided among the participating patient, the clinician, and the clinic? Further, how does one determine the potential additional costs to participants arising from asking a sensitive question or doing a clinical examination? Some participants in our study suggested that consent for participation in PACER-type research could be done prior to the clinic visit. Would patients receive compensation for the extra time dedicated to this early consent process? Finally, with all costs accounted for, would PACER be more cost effective than other research methods?

Other key domains described by participants were the impact of PACER on people (both patients and clinicians) and organizations, and on the patient-clinician relationship. Here, the ethical principles of beneficence—referring to benefits to the patient and clinician—and nonmaleficence—referring to potential harms to the patient and clinician—come to the fore. Participants noted that PACER could improve the quality of research, generating societal benefits to patient, clinicians, and communities. On the one hand, participants noted PACER could lead to better patient care and clinician knowledge. Higher quality research should eventually result in improved patient care by providing physicians with more accurate and detailed information, better therapeutics, etc. And clinicians' potentially learning more about their patients' health via research participation could be beneficial for both parties. However, those benefits need to be weighed against the negatives described, such as additional time burdens to patients and clinicians, more clinician responsibility, and impacts to clinic flow.

Similarly, participants noted the potential for PACER to both enhance and diminish the patient-clinician relationship. As previously discussed, some patients' involvement could be heavily influenced by the participation of their regular clinician. If patients see that their

clinicians are invited to and are participating in research, some may be intrigued by or pleased with their clinician's association and attempt to include them. However, others—especially members of patient populations who have historically been marginalized or mistreated by researchers—may resent the inclusion of research into their clinic visit. This may lead to a widening gap of mistrust between clinicians and patients. A physician's oath is to do no harm and to provide the best care possible; they have an obligation to act in accordance with their best judgment of their patients' needs and to maintain a trusting connection with patients. Doing so may be particularly challenging in certain settings or with certain populations. Study participation should not come at the cost of this relationship and should avoid compromising these ethical principles. With the increased use of pragmatic clinical trials and the focus on learning health systems, the impacts of embedding research into clinical care for both clinicians and patients are receiving more attention. Considering the impacts of PACER on patients, clinicians, and the relationship between patient and physician will be essential in future work.

One major limitation of this study is the significant difficulty in imagining PACER implementation and its impacts without a demonstration or simulation. Without more direct experience with PACER, participant responses to our use case may over- or underestimate PACER's potential positive and negative impacts. Therefore, a next step for attaining empirical data about the ethics of PACER implementation could involve creating videos that demonstrate different PACER implementation scenarios. These videos could illustrate various approaches to consent, use different clinic staff (e.g., physicians, nurses, etc.) to inform participants about PACER, demonstrate what it would be like to ask a simple research question or perform an additional exam, and show how a study question may prompt a need for follow-up. As a video cannot convey all potential nuances of implementing PACER in the real-world setting, a subsequent step would be to pilot different implementation strategies in real clinics. Such pilot studies should obtain input from participating patients, clinicians, clinic staff, and clinic administrative leaders. Patient-related insights should include their views on the consent process, understanding of the research and their clinician's role in the research, the impact of PACER on clinician trust and the clinician-patient relationship, and compensation. Clinician insight should cover the perceived burden; impact on the clinic visit, clinic flow, and clinician-patient relationship; opportunities for compensation; and overall satisfaction. Staff and administrator input should address the impact on clinic flow, clinic visit time, cost, and opportunities for compensation.

We acknowledge additional study limitations. We did not consider PACER implementation with pediatric patient participants because of the unique considerations related to involving children. We present information related to only financial compensation because of inconsistencies in asking about non-financial compensation. Additionally, we acknowledge that we may have failed to capture the full gamut of possible viewpoints as some participants expressed differing or contradictory opinions within one interview and certain demographic groups were insufficiently represented in this preliminary study. Future work should consider PACER with the pediatric population, as well as alternatives to financial compensation for patients, clinicians, and clinics (e.g., such as authorship on a manuscript for clinicians or access to a health-related support application for patients). The participants of this study were largely from the Chicagoland area and overrepresented patients (45% compared to 26% and 30% of research experts/bioethicists and clinic staff, respectively) and women (66%), potentially limiting generalizability [23]. This was the initial exploration of the PACER concept, and so we limited the participation until saturation was reached. Our investigation is limited because of our sample size and geographic location. In choosing to focus on a diverse population from one of the nation's largest cities, we hope this will provide a generalizable foundation on which much-needed, further feasibility studies may be based.

## Conclusion

This study describes input from patients, clinicians, clinic staff, research experts, and bioethicists on potential ethical, sociotechnical, and implementation issues raised by PACER, a novel approach to conducting efficient, scalable, and cost-effective minimal-risk research in the clinic setting that leverages emerging healthcare computing technologies. We identified five key domains: impact on clinical research, consent, compensation, impacts on people and organizations, and implementation. These results highlight the need for further empirical study of the ethical issues associated with implementing PACER-type research.

## Supporting information

**S1 File. Interview Guide.**
(PDF)

**S1 Fig. Zipf analysis.** "All Codes" refers to topic codes in addition to the subcodes related to each topic. "Leaf Codes" include only the subcodes related to each topic.
(TIF)

**S2 Fig. Choice of consent type by participant Type (n = 47).** Asterisk indicates responses include those coded as "depends" and anyone who endorsed more than one response. One person did not endorse any of the consent types presented.
(TIF)

**S3 Fig. Choice of consent type by gender Type (n = 47).** Single asterisk indicates responses include those coded as "depends" and anyone who endorsed more than one response. Double asterisk indicates data includes one person who self-identified as non-binary. One person did not endorse any of the consent types presented.
(TIF)

## Acknowledgments

The authors have no acknowledgements.

## Author contributions

**Conceptualization:** Kelly Nicole Michelson, Justin Starren.

**Data curation:** Emma Friedman, Shruti Seghal.

**Formal analysis:** Emma Friedman, Kelly Nicole Michelson, Shruti Seghal, Russell Steans, Matthew J. Baumann.

**Funding acquisition:** Kelly Nicole Michelson, Justin Starren.

**Methodology:** Emma Friedman, Kelly Nicole Michelson, Shruti Seghal, Mohammad Hosseini, Matthew J. Baumann, Amanda K. Venables, Theresa L. Walunas, Justin Starren.

**Project administration:** Emma Friedman, Shruti Seghal.

**Supervision:** Kelly Nicole Michelson, Justin Starren.

**Validation:** Emma Friedman, Shruti Seghal, Mohammad Hosseini, Matthew J. Baumann, Amanda K. Venables, Theresa L. Walunas, Justin Starren.

**Visualization:** Emma Friedman, Kelly Nicole Michelson, Shruti Seghal, Russell Steans, Matthew J. Baumann, Amanda K. Venables, Theresa L. Walunas, Justin Starren.

**Writing – original draft:** Emma Friedman, Kelly Nicole Michelson.

**Writing – review & editing:** Shruti Seghal, Russell Steans, Mohammad Hosseini, Matthew J. Baumann, Amanda K. Venables, Theresa L. Walunas, Justin Starren.

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
