## [Decision Letter · Decision Letter 0]

14 Aug 2024

PONE-D-24-21081Implications for Precision Accelerated Clinically Embedded Research (PACER): a novel technology-enabled approach to conducting minimal-risk research in outpatient community healthcare settingsPLOS ONE

Dear Dr. Michelson,

Thank you for submitting your manuscript to PLOS ONE. After careful consideration, we feel that it has merit but does not fully meet PLOS ONE’s publication criteria as it currently stands. Therefore, we invite you to submit a revised version of the manuscript that addresses the points raised during the review process.

We look forward to receiving your revised manuscript.

Kind regards,

Roberto Scendoni

Academic Editor

PLOS ONE

Journal Requirements:

Research reported in this publication was supported, by The Greenwall Foundation and the National Institutes of Health's National Center for Advancing Translational Sciences, Grant Number UL1TR001422. The content is solely the responsibility of the authors and does not necessarily represent the official views of the Greenwall Foundation or the National Institutes of Health.

Author who received award: Justin Starren

Funder: The Greenwall Foundation

URL: https://greenwall.org/

The funder did not play any role in the study design, data collection and analysis, decision to publish, or preparation of the manuscript.

Author who received funding: Justin Starren

Grant Number: UL1TR001422

Name of Funder: National Institutes of Health's National Center for Advancing Translational Sciences

The funder did not play any role in the study design, data collection and analysis, decision to publish, or preparation of the manuscript

I have read the journal's policy and the authors of this manuscript have the following competing interests:

Dr. Kelly Michelson has received funding for other work from the Patient Centered Outcomes Research Institute and the National Institutes of Health

Mohammad Hosseini was supported by the National Institutes of Health’s National Center for Advancing Translational Sciences (UL1TR001422)

Matthew Baumann's employment is supported by a grant from the National Evaluation Center of the Network of the National Library of Medicine (NEC/NNLM)

We note that one or more of the authors are employed by a commercial company. 

“The funder provided support in the form of salaries for authors, but did not have any additional role in the study design, data collection and analysis, decision to publish, or preparation of the manuscript. The specific roles of these authors are articulated in the ‘author contributions’ section.”

4. In this instance it seems there may be acceptable restrictions in place that prevent the public sharing of your minimal data. However, in line with our goal of ensuring long-term data availability to all interested researchers, PLOS’ Data Policy states that authors cannot be the sole named individuals responsible for ensuring data access (http://journals.plos.org/plosone/s/data-availability#loc-acceptable-data-sharing-methods).

Reviewers' comments:

Reviewer's Responses to Questions

**Comments to the Author**

1. Is the manuscript technically sound, and do the data support the conclusions?

Reviewer #1: Partly

Reviewer #2: Yes

2. Has the statistical analysis been performed appropriately and rigorously? 

Reviewer #1: Yes

Reviewer #2: N/A

3. Have the authors made all data underlying the findings in their manuscript fully available?

Reviewer #1: Yes

Reviewer #2: Yes

4. Is the manuscript presented in an intelligible fashion and written in standard English?

Reviewer #1: Yes

Reviewer #2: Yes

5. Review Comments to the Author

**Reviewer #1:**  Review

• The introduction needs to be revised as it is unclear. The discussion on PACER is not adequately explained—what it is and what it entails are not clear without referring to bibliographic entry "7," which makes the paper difficult to read. Consequently, the authors present isolated statements about the system, which make it difficult for readers to understand what this system entails. Moreover, although the authors extol the system, these statements would benefit from better justification or even removal.

• Were the participants adequately prepared before administering the questionnaire? Were they informed, even in broad terms, about the issues at hand? Was any information provided on this topic?

• Please provide a better justification for the following statement: "However, obtaining written informed consent from participating patients could prolong the clinic encounter—reducing efficiency and potentially affecting participation" (line 483).

• The results are not well aligned with the description. Given the description of the participants, were differences in viewpoints considered?

• Overall, I find it challenging to understand the utility of the work, and the target audience of the article is not clear.

**Reviewer #2:**  Thank you for the oprtonuity to review this paper that qualitatively seeks to understand the ramifications of a distributive methodology to conduct low risk (human subject risk) research. This process may help in increasing equitable burden of research while still holding to the justice principles.

Overall this was a well done description of interview methodology and provides a clear path to data collection and distillation.

the only methodological concern i had was i was not clear in how the interviews were conducted, ie were they individual or group settings. if in groups, were the groups mixed with respect to the types of participants (ie ethicists, providers, regulatory)

I also have a generalizability concern, subjects were restricted to the Chicago area. This concern is addressed in the limitations discussion, however, authors admit that the participants mirrored the Chicago mixture of society, but that does not necessarily mirror the greater American population. I would ask that authors make this limitation a little stronger instead of minimizing this area.

overall this is a well done and well written report. Thank you for your contribution.

6. PLOS authors have the option to publish the peer review history of their article (what does this mean? ). If published, this will include your full peer review and any attached files.

**Do you want your identity to be public for this peer review?** For information about this choice, including consent withdrawal, please see our Privacy Policy .

Reviewer #1: No

Reviewer #2: **Yes: ** David Wampler, PhD, LP, FAEMS

---

## [Author Response · Author response to Decision Letter 1]

22 Oct 2024

Part I: Reviewer Comments and Responses

Each reviewer comment is numbered. Our responses are below each comment, and all new and existing excerpts from the text are in quotation marks. All page and line numbers correspond to the new, edited version of the manuscript.

Reviewer #1:

(1) The introduction needs to be revised as it is unclear. The discussion on PACER is not adequately explained—what it is and what it entails are not clear without referring to bibliographic entry "7," which makes the paper difficult to read. Consequently, the authors present isolated statements about the system, which make it difficult for readers to understand what this system entails.

We thank you for your comment requesting additional details about the functioning and technical aspects of PACER, citing bibliographic entry 7. To that end, we have made the following additions to the introduction section of the manuscript. We explain in additional detail related to what we hope PACER can achieve and how we envision the PACER architecture would function. Please see the manuscript pages 3-4 lines 47-74 and below and we have added the appropriate reference for this citation:

"PACER’s mission is to maximize opportunities for broad and diverse participation in clinical research by expanding access to research studies and thereby improving the generalizability of results. This goal is shared by the National Institutes of Health, among other research organizations. The NIH’s newly appointed director has stated, “If we can build and support a research infra- structure that seamlessly integrates innovative science with routine clinical care, this will not only help us deliver high-quality results for people in a less resource-intensive way, but will allow us to get into diverse communities where we otherwise haven’t been.”[23] This aim is one our research team shares, and PACER is an embodiment of this aspiration. And with the increasing role of EHRs in clinical research, this new research infrastructure must interface with EHRs in non-AMC-affiliated and under-resourced clinics.

PACER leverages recent advances in technology called CDS hooks [11] to enable researchers to create and deliver to clinicians very short (under two minutes) research prompts—such as asking a question or doing a brief exam—into the clinical encounter. In doing so, PACER seeks to enable high-speed, real-time, scalable, minimal-risk research through rapid bi-directional transmission of information between a central hub and many distributed electronic health records (EHRs). This distributed approach allows research prompts to be sent to any participating clinician across the country, regardless of their location or which EHR they use. The responses to the prompts (e.g., answers to questions or results of the medical exam) would be documented in the medical record, thereby providing data about a symptom not typically asked about or physical finding not regularly included in a clinic visit. This additional information could also be returned to the central hub to rapidly answer research questions. The following example about how PACER might have been useful during the recent COVID-19 pandemic further explains PACER and its potential impact. After clinicians noted that some patients with COVID-19 experienced anosmia, PACER could have sent prompts to clinicians around the country whose patients had symptoms suggestive of COVID-19 requesting that the clinician inquire about anosmia. This information could have been paired with COVID-19 test results to rapidly confirm or deny an association between anosmia and COVID-19."

(2) Moreover, although the authors extol the system, these statements would benefit from better justification or even removal.

Thank you for this important reminder to temper our statements. Throughout the paper, we are careful to use terms such as “could,” “hope,” “seek,” or “maybe.” We have edited the manuscript to moderate our statements and provide a more balanced perspective, and we note PACER’s limitations.

Below, we list a few examples of how we moderated our statements.

1. On page 4 line 87, we deleted “enormous” regarding PACER’s potential.

2. On page 22 line 589, we changed “even” to “might” in the context of PACER’s potential to support improved resource allocation.

3. On page 32 line 854, we have removed the modifier, “enthusiastically” to describe our participants’ responses to PACER’s potential benefits.

(3) Were the participants adequately prepared before administering the questionnaire? Were they informed, even in broad terms, about the issues at hand? Was any information provided on this topic?

Thank you for these questions—participant preparation is an important issue that we appreciate clarifying. Participants were informed in broad terms of the subject of the study and the most salient features of the discussion through an information sheet provided to potential participants prior to interviews. Additionally, at the commencement of the interview, the interviewer gave an overview of PACER to contextualize the discussion. We added clarifying text on pages 7-8 lines 287-297 and here it is below:

"Prior to confirming their involvement in the study, potential interviewees received an information sheet, sent electronically via email. The information sheet included a brief description PACER, of the study’s aims, the risks and benefits of participation, and study and institutional contact information. When the interviews commenced, the interviewer gave a brief description of PACER and how PACER would function and offered the participants the opportunity to ask their own questions prior to the interview questions."

(4) Please provide a better justification for the following statement: "However, obtaining written informed consent from participating patients could prolong the clinic encounter—reducing efficiency and potentially affecting participation" (line 483).

We apologize for the lack of clarity. The issue of consent, and how the consent process would interact with clinical workflow, is important. As is typical in surveys of research subject attitudes about consent, there was a heterogeneity of opinion. In the results we noted,

“Some comments reflected the concern that truncating the consent process (e.g., requiring only oral permission) risks harming the process’s integrity and inhibiting patients from making well-informed choices. However other participants described how obtaining informed consent has its own associated burden” (page 29 line 761-765).

In addition, in Table 2, Domain 2 Consent, Subtheme 1b Clinic factors, we report the quote,

“A signed consent form adds time, complexity, interruption ...if the written information could be passed off like in the MA check right, and just say, ‘Hey, we're doing a little—we're doing some research on a medication you're taking. Here's just some information, right if you're comfortable, the provider may ask some questions, and you don't have to say, if you want to, and here's a little bit of information you can read while you wait for the provider to come into the room’ that's pretty fluid” (12).

To further clarify this topic, we have changed the sentence to the following on page 29 lines 764-765.

"However, other participants noted that obtaining written informed consent could prolong the clinical encounter—reducing efficiency and potentially affecting participation."

(5) The results are not well aligned with the description. Given the description of the participants, were differences in viewpoints considered?

In retrospect, we realize that the opening of the discussion section may not have captured the overall sense of the results. We have reworked this section including adding several clarifying statements including on page 28 at line 727-735:

"Most reactions to PACER were positive. Participants indicated that PACER could improve clinical research processes and increase research participation, as well as advancing scientific knowledge. At the same time, the results demonstrated diverse views about implementing various aspects of PACER methodology. We discovered a diversity of opinions within each group of respondents. Although the responses were not amenable to a formal statistical analysis, we did not identify obvious trends within or among each category of interviewee. Thus, there was no unanimity regarding a single optimal design for a PACER implementation, and future implementers will need to weigh the various opinions."

We also added at page 35 lines 908-911:

"Additionally, we acknowledge that we may have failed to capture the full gamut of possible viewpoints as some participants expressed differing or contradictory opinions within one interview and certain demographic groups were insufficiently represented in this preliminary study."

(6) Overall, I find it challenging to understand the utility of the work, and the target audience of the article is not clear.

Thank you for bringing up the important issues of the utility of this work and our target audience. The primary goals of PACER are to increase the pace of and expand the access to biomedical research. In this particular work, we are exploring our interviewees’ viewpoints on a number of topics related to PACER’s design and implementation. To better describe the utility of the work, we have added the following to the introduction on page 5 at lines 217-219:

"By exploring and describing diverse viewpoints related to PACER, and by identifying potential challenges imposed by PACER a priori, we intend for this work to inform the creation of guidelines for future PACER implementation."

We feel these data will inform researchers interested in implementing PACER and clinicians and patients who may be involved in research that uses PACER methodology. To further clarify this we have included the following to the end of the first paragraph in the discussion on page 28-29 at lines 744-756.

"We believe that this study of participants’ perceptions of PACER and its ethical implications, as well as how to implement PACER, can help guide the wider research and bioethics communities—including researchers, bioethicists, institutional review boards (IRBs), and others—who may wish to implement PACER."

Reviewer #2:

(1) Thank you for the opportunity to review this paper that qualitatively seeks to understand the ramifications of a distributive methodology to conduct low risk (human subject risk) research. This process may help in increasing equitable burden of research while still holding to the justice principles. Overall this was a well done description of interview methodology and provides a clear path to data collection and distillation.

Thank you for these positive comments; we appreciate your vote of confidence and your sharing our vision.

(2) The only methodological concern I had was I was not clear in how the interviews were conducted, e.g. were they individual or group settings. if in groups, were the groups mixed with respect to the types of participants (e.g. ethicists, providers, regulatory)

Thank you so much for bringing this clarification question to our attention. All interviews were conducted individually (i.e., one-on-one) via teleconferencing (over Zoom). To make this clearer, we have added the following phrase--”one-on-one"to the manuscript—please see page 6 line 191 and also below:

"We conducted one-on-one, qualitative semi-structured interviews with patients, clinicians, research experts, and bioethicists between February and August 2022."

(3) I also have a generalizability concern, subjects were restricted to the Chicago area. This concern is addressed in the limitations discussion, however, authors admit that the participants mirrored the Chicago mixture of society, but that does not necessarily mirror the greater American population. I would ask that authors make this limitation a little stronger instead of minimizing this area.

Generalizability is one of our top priorities, so we thank you for your comment on this subject. We more strongly acknowledge this limitation below, along with explaining our reasoning (that we hope that Chicago is a good starting off point on which to base future endeavors and that saturation from our Zipf analysis was reached). Please see page 36 lines 875-879 and below:

"Given that Chicagoland is a highly diverse region, we felt it a good place to start this exploration of people’s view about PACER. Even so, it is clear that future studies with larger geographic coverage, and inclusion of ethnic groups not well represented in Chicagoland, e.g. Native American, Alaska Native, Native Hawaiian, and Pacific Islander, are needed."

(4) Overall this is a well done and well written report. Thank you for your contribution.

Thank you. We so appreciate the opportunity to resubmit this manuscript and feel it is much improved as a result of the requested revisions.

Part II: Editor Comments and Responses

The journal’s editor comments are listed below, and each comment is followed by a response. They are numbered for convenience, and the first sentence of each comment is bolded.

Journal Requirements:

Thank you so much for sharing these helpful resources with us. We are very grateful to have this information to consult. To that end, we have done the following:

1. We have deleted the address listed for the corresponding author and just kept the email address as instructed.

2. Headings: In keeping with sentence capitalization, we changed the “m” in “methods” in the heading in line 121 from uppercase to lowercase as the guide requires. On the same page at line 131, we changed the “g” in “Interview guide” from uppercase to lowercase. On the following page (at line 164), we changed from uppercase to lowercase several letters in the heading, “Participant recruitment and data collection.” We changed the “a” in the heading, “Data analysis” at line 194 from uppercase to lowercase. At line 261, we changed the heading “Impact on clinical research” to lowercase from uppercase. We also changed the “t” in type from uppercase to lowercase in heading “Consent type” at line 285. At line 321, we changed the “c” in “Consent” from uppercase to lowercase. At line 335, we changed the uppercase “P” to lowercase. At line 377, there is a heading with a slash in it. We changed the “c” in compensation from uppercase to lowercase, but we’re unsure if we are be permitted to keep the slash or the capitalization next to it. We also changed the “c” in compensation from uppercase to lowercase at lines 377 and 395. At line 409, we changed from uppercase to lowercase the “p” and “o” in “Impacts on people and organizations.” The “c” in “considerations” was changed from uppercase to lowercase at line 480. We also changed to lowercase letters in the heading on line 501, “Post-study contact.” The “q” in question at line 535 is also now lowercase. At line 549, the “t” in “topics” was changed to lowercase from uppercase.

We also changed the headings in our tables to match this style.

Research reported in this publication was supported, by The Greenwall Foundation and the National Institutes of Health's National Center for Advancing Translational Sciences, Grant Number UL1TR001422. The content is solely the responsibility of the authors and does not necessarily represent the official views of the Greenwall Foundation or the National Institutes of Health.

We note that you have provided funding information that is not currently declared in your Funding Statement. However, funding information should not appear in the Acknowledgments section or other areas of your manuscript. We will only publish funding information present in the Fundin

---

## [Decision Letter · Decision Letter 1]

1 Dec 2024

PONE-D-24-21081R1Implications for Precision Accelerated Clinically Embedded Research (PACER): a novel technology-enabled approach to conducting minimal-risk research in outpatient community healthcare settingsPLOS ONE

Dear Dr. Michelson,

Thank you for submitting your manuscript to PLOS ONE. After careful consideration, we feel that it has merit but does not fully meet PLOS ONE’s publication criteria as it currently stands. Therefore, we invite you to submit a revised version of the manuscript that addresses the points raised during the review process.

We look forward to receiving your revised manuscript.

Kind regards,

Roberto Scendoni

Academic Editor

PLOS ONE

**Journal Requirements:**

Reviewers' comments:

Reviewer's Responses to Questions

**Comments to the Author**

1. If the authors have adequately addressed your comments raised in a previous round of review and you feel that this manuscript is now acceptable for publication, you may indicate that here to bypass the “Comments to the Author” section, enter your conflict of interest statement in the “Confidential to Editor” section, and submit your "Accept" recommendation.

Reviewer #1: All comments have been addressed

Reviewer #3: All comments have been addressed

2. Is the manuscript technically sound, and do the data support the conclusions?

Reviewer #1: Yes

Reviewer #3: Yes

3. Has the statistical analysis been performed appropriately and rigorously? 

Reviewer #1: N/A

Reviewer #3: N/A

4. Have the authors made all data underlying the findings in their manuscript fully available?

Reviewer #1: Yes

Reviewer #3: Yes

5. Is the manuscript presented in an intelligible fashion and written in standard English?

Reviewer #1: Yes

Reviewer #3: Yes

6. Review Comments to the Author

**Reviewer #1: ** Overall, the revisions result in a significant improvement of the manuscript. However, it is suggested to include some references in the text concerning the ethical section.

De Micco F, Scendoni R. Three Different Currents of Thought to Conceive Justice: Legal, and Medical Ethics Reflections. Philosophies. 2024; 9(3):61. https://doi.org/10.3390/philosophies9030061

Van Biesen W, Decruyenaere J, Sideri K, Cockbain J, Sterckx S. Remote digital monitoring of medication intake: methodological, medical, ethical and legal reflections. Acta Clin Belg. 2021 Jun;76(3):209-216. doi: 10.1080/17843286.2019.1708152.

**Reviewer #3:**  It has been my pleasure to review this excellent paper, which qualitatively examines the ramifications of a distributive methodology to conduct low-risk (human subject risk) research. It is my hope that this process will help increase equitable burdens of research while at the same time adhering to justice principles. As a whole, this was a well-written description of interview methodology and provided a clear path for collecting and distilling data.

7. PLOS authors have the option to publish the peer review history of their article (what does this mean? ). If published, this will include your full peer review and any attached files.

**Do you want your identity to be public for this peer review?** For information about this choice, including consent withdrawal, please see our Privacy Policy .

Reviewer #1: No

Reviewer #3: **Yes: ** Ola Mousa

---

## [Author Response · Author response to Decision Letter 2]

16 Dec 2024

Author Revisions and Replies

Reviewer's Responses to Questions and Comments to the Author

1. If the authors have adequately addressed your comments raised in a previous round of review and you feel that this manuscript is now acceptable for publication, you may indicate that here to bypass the “Comments to the Author” section, enter your conflict of interest statement in the “Confidential to Editor” section, and submit your "Accept" recommendation.

Reviewer #1: All comments have been addressed

Reviewer #3: All comments have been addressed

We thank both Reviewer #1 and Reviewer #3 for this vote of confidence in our work and revisions.

2. Is the manuscript technically sound, and do the data support the conclusions?

Reviewer #1: Yes

Reviewer #3: Yes

We thank both Reviewer #1 and Reviewer #3 for their affirmation of this point.

3. Has the statistical analysis been performed appropriately and rigorously?

Reviewer #1: N/A

Reviewer #3: N/A

4. Have the authors made all data underlying the findings in their manuscript fully available?

Reviewer #1: Yes

Reviewer #3: Yes

We are making these data available, submitting our data to deposit in the repository ICPSR. Upon acceptance of our manuscript for publication, we will provide you with the detailed information of our ICPSR deposit, including a DOI, which is currently in progress.

5. Is the manuscript presented in an intelligible fashion and written in standard English?

Reviewer #1: Yes

Reviewer #3: Yes

6. Review Comments to the Author

Reviewer #1: Overall, the revisions result in a significant improvement of the manuscript. However, it is suggested to include some references in the text concerning the ethical section.

De Micco F, Scendoni R. Three Different Currents of Thought to Conceive Justice: Legal, and Medical Ethics Reflections. Philosophies. 2024; 9(3):61. https://doi.org/10.3390/philosophies9030061

Van Biesen W, Decruyenaere J, Sideri K, Cockbain J, Sterckx S. Remote digital monitoring of medication intake: methodological, medical, ethical and legal reflections. Acta Clin Belg. 2021 Jun;76(3):209-216. doi: 10.1080/17843286.2019.1708152.

We thank Reviewer #1 for the kind words regarding the improvement of the manuscript and the acknowledgement of our previous efforts.

We also thank them for their suggestion to add references to the text, specifically in the ethical section and for pointing out these interesting articles. The Van Biesen, et al. article focuses primarily on the use of digital pills in clinical practice. The discussion of research mentions that the pill could be used in clinical trials and that FDA approval was based on non-randomized trials. The paper did not directly address potential bioethical issues related to point-of-care research as we describe, and it focuses exclusively on a particular intervention. Therefore, we feel that it is not applicable to our article.

The paper by De Micco and Scendoni outlined the concept of justice, providing definitions in the broadest ethical terms. However, the article did not discuss justice in a specifically bioethical context. Since our area of investigation is bioethics, that context is crucial. Additionally, in our manuscript, justice is discussed in the context of compensation for research participation and includes a reference which more directly pertains to the comments made. Therefore, we have chosen not to include the suggested article.

We did remove a repetitive sentence that emphasized justice starting at line 563 on page 30, “Thus an important aspect of patient compensation relates to bioethical principles justice and fairness.”

Reviewer #3: It has been my pleasure to review this excellent paper, which qualitatively examines the ramifications of a distributive methodology to conduct low-risk (human subject risk) research. It is my hope that this process will help increase equitable burdens of research while at the same time adhering to justice principles. As a whole, this was a well-written description of interview methodology and provided a clear path for collecting and distilling data.

We thank Reviewer #3 so much for these words of support. We are so pleased to have earned that approval.

7. PLOS authors have the option to publish the peer review history of their article (what does this mean?). If published, this will include your full peer review and any attached files.

Do you want your identity to be public for this peer review? For information about this choice, including consent withdrawal, please see our Privacy Policy.

Reviewer #1: No

Reviewer #3: Yes: Ola Mousa

---

## [Editor Report · Decision Letter 2]

17 Jan 2025

Implications for Precision Accelerated Clinically Embedded Research (PACER): a novel technology-enabled approach to conducting minimal-risk research in outpatient community healthcare settings

PONE-D-24-21081R2

Dear Dr. Michelson,

We’re pleased to inform you that your manuscript has been judged scientifically suitable for publication and will be formally accepted for publication once it meets all outstanding technical requirements.

Kind regards,

Emily Lund

Academic Editor

PLOS ONE

Additional Editor Comments (optional):

The authors have adequately addressed all prior comments in their revision.
---

## [Editor Report · Acceptance letter]

PONE-D-24-21081R2

PLOS ONE

Dear Dr. Michelson,

I'm pleased to inform you that your manuscript has been deemed suitable for publication in PLOS ONE. Congratulations! Your manuscript is now being handed over to our production team.

Kind regards,

on behalf of

Dr. Emily Lund

Academic Editor

PLOS ONE